# Free-Radical-Mediated Formation of Trans-Cardiolipin Isomers, Analytical Approaches for Lipidomics and Consequences of the Structural Organization of Membranes [note 1]

**DOI:** 10.3390/biom10081189

**Published:** 2020-08-15

**Authors:** Fabrizio Vetica, Anna Sansone, Cesare Meliota, Gessica Batani, Marinella Roberti, Chryssostomos Chatgilialoglu, Carla Ferreri

**Affiliations:** 1Istituto per la Sintesi Organica e la Fotoreattività, Consiglio Nazionale delle Ricerche, Via Piero Gobetti 101, 40129 Bologna, Italy; fabrizio.vetica@isof.cnr.it (F.V.); anna.sansone@isof.cnr.it (A.S.); cesare.meliota@studio.unibo.it (C.M.); gessica.batani@studio.unibo.it (G.B.); 2Department of Pharmacy and Biotechnology, University of Bologna, Via Belmeloro 6, 40126 Bologna, Italy; marinella.roberti@unibo.it; 3Center for Advanced Technologies, Adam Mickiewicz University, 61-614 Poznań, Poland

**Keywords:** cardiolipin, linoleic acid, free radicals, cis-trans isomerization, γ-irradiation, Fenton reaction, liposome dimension

## Abstract

Free-radical-mediated processes, such as peroxidation, isomerization and hydrogenation affecting fatty acid integrity and biological functions, have a trans-disciplinary relevance. Cardiolipins (CL, (1,3-diphosphatidyl-*sn*-glycerol)) and tetra-linoleoyl-CL are complex phospholipids, exclusively present in the Inner Mitochondrial Membrane (IMM) lipids, where they maintain membrane integrity and regulate enzyme functionalities. Peroxidation pathways and fatty acid remodeling are known causes of mitochondrial disfunctions and pathologies, including cancer. Free-radical-mediated isomerization with the change of the cis CL into geometrical trans isomers is an unknown process with possible consequences on the supramolecular membrane lipid organization. Here, the formation of mono-trans CL (MT-CL) and other trans CL isomers (T-CL) is reported using CL from bovine heart mitochondria and thiyl radicals generated by UV-photolysis from 2-mercaptoethanol. Analytical approaches for CL isomer separation and identification via ^1^H/^13^C NMR are provided, together with the chemical study of CL derivatization to fatty acid methyl esters (FAME), useful for lipidomics and metabolomics research. Kinetics information of the radical chain isomerization process was obtained using γ-irradiation conditions. The CL isomerization affected the structural organization of membranes, as tested by the reduction in unilamellar liposome diameter, and accompanied the well-known process of oxidative consumption induced by Fenton reagents. These results highlight a potential new molecular modification pathway of mitochondrial lipids with wide applications to membrane functions and biological consequences.

## 1. Introduction

In lipid chemistry, the free-radical-mediated cis-trans isomerization process contributes, together with hydrogenation and lipid peroxidation processes, to cause alteration of the naturally occurring unsaturated fatty-acid-containing lipids [1,2,3]. These processes have a trans-disciplinary relevance, being identified in biological pathways of radical stress as well as in industrial food and nutraceutical production [3,4,5,6]. Analytical protocols have been established to determine trans lipid presence in living organisms as well as in nutraceutical formulations, which can affect health [4,5,6].

Cardiolipin (CL) is a unique phospholipid, which is localized and synthesized in the inner mitochondrial membrane (IMM), where it represents a fairly high percentage of the total lipids (about 25% of the total lipid composition of bovine heart mitochondria) [7,8,9]. It is characterized by a bi-phospholipid structure, bridged by an additional glycerol moiety, and four fatty acid residues (Figure 1). This distinctive dimeric structure has a conical shape, therefore imparting a peculiar contribution in the formation of lipid bilayers, causing the extensive folding of the IMM cristae [10]. The type of the CL fatty acid residues is intriguing from the chemical biology point of view. The immature CL, containing mainly palmitic (16:0) and oleic (9cis-18:1) acids, is remodeled by de-acylation and trans-acylation reactions affording the final composition of mature CL, containing predominantly polyunsaturated fatty acids (PUFAs) with linoleic acid (9cis,12cis-18:2) as the most abundant residue [11,12]. For instance, bovine heart cardiolipin contains 95% of linoleic acid, while the remaining 5% is constituted by vaccenic acid (11cis-18:1) and a few other saturated and unsaturated FAs [13]. The enrichment of PUFAs such as linoleic acid is well known to be crucial for the lipid organization of the IMM and binding with the complexes I, III, IV, V, of the respiratory chain, also stabilizing the super complexes [14].

The structural contribution of CL is paralleled by a very important chemical contribution in the frame of the mitochondrial free radical reactivity, particularly focusing on the reactivity of tetra-linoleoyl-CL (Figure 1). The bisallylic H-atoms of the linoleoyl residues can be abstracted by an oxygen-centered radical (for example, a peroxyl radical generated under the conditions of mitochondrial stress). Subsequent reaction with molecular oxygen and the generation of a lipid peroxyl radical activate the chain process, which is responsible for the intrinsic apoptotic pathway with release of cytochrome c [15].

We were interested by the fact that double bond isomerization was never reported for CL, although such a process cannot be ruled out if one considers the mechanism of thiyl radical-catalyzed reversible addition to lipids (**L**); it occurs with the radical mechanism shown in Figure 2a, consisting of a reversible addition of thiyl radical (RS^•^) to the double bond forming the trans geometry as the most thermodynamically favorable structure. It is worth noting that (i) the radical RS^•^ acts as a catalyst for cis-trans isomerization, and (ii) positional isomers cannot be reaction products because the mechanism does not allow a double bond shift [2,16,17]. Considering polyunsaturated substrates, the isomerization mechanism occurs as a step-by-step process, i.e., each isolated double bond behaves independently (Figure 2b) [18,19]. With linoleic-acid-containing liposomes and micelles, we demonstrated that the mono-trans isomers (**L_EZ_**, **L_ZE_**) are the first formed products and the competition with peroxidation occurs under oxygen conditions [20]. Moreover, we examined kinetics and products of the thiyl radical reaction with methyl linoleate, thus completing the picture of PUFA reactivity [19]. This knowledge supports the feasibility of cis-trans CL isomerization by free radical stress that could occur in the specific mitochondrial environment. So far, this transformation is unknown.

The analytical aspect concerning the identification of trans CL isomers is a crucial issue to address, with the specific goal to have protocols that can examine biological specimens. At present, CL analysis includes: (i) the isolation of CL via preparative thin layer chromatography (TLC) and transesterification to obtain the corresponding fatty acid methyl esters (FAME) under acidic conditions (HCl, H_2_SO_4_ or BF_3_ in methanol) to proceed with the gas chromatographic (GC) identification of the fatty acid residues [21,22,23,24]; (ii) HPLC with tandem MS/MS analysis to identify cardiolipins among other lipid classes [25,26,27,28,29]. Based on our experience in fatty acid-based lipidomics [2,19], the transesterification conditions must ensure a quantitative performance and the analytical conditions need to separate and identify cis and trans isomers. From the biological point of view, it is worth underlining that fatty acid double bonds are generated stereo- and regio-specifically by desaturase enzymes [30], and only the cis geometry provides the known membrane asset and signaling properties [31,32]. The permeability and fluidity of the membranes are profoundly affected by the cis-trans isomerization of unsaturated fatty acids, as demonstrated for membrane biophysical and dimensional properties using cis and trans monounsaturated fatty acid (MUFA) residues [33,34,35]. We also showed that conditions of cellular stress cause the degradation of sulfur-containing proteins and the formation of diffusible thiyl radicals able to induce cis−trans isomerization in unsaturated lipid vesicles [2,36].

In the frame of our research on mono-trans PUFA isomers as biomarkers to be used to detect isomerization in cells [37,38], animals [39,40] and humans [41,42,43], we wish to present our results on the identification and characterization of trans CL isomers by thiyl radical-catalyzed cis-trans isomerization of bovine hearth CL employing photolysis and γ-radiolysis techniques. Furthermore, we developed the analytical protocol for the recognition of trans CL via ^1^H and ^13^C NMR and gave a fully calibrated procedure for CL transesterification into their corresponding FAME for GC analysis. We also performed liposome experiments to mimic trans CL formation in membranes addressing the influence of geometrical isomerism on the compartment formation process and the relationship with oxidative processes.

## 2. Materials and Methods

Unless otherwise noted, all commercially available compounds were used without further purification. Water was purified with a Millipore system. Bovine hearth cardiolipin and 2-oleoyl-1-palmitoyl-*sn*-glycero-3-phosphocholine (POPC) were purchased from Larodan Inc (Solna, Sweden); Ferrous ammonium sulfate Fe(NH_4_)_2_(SO_4_)_2_ × 6H_2_O (Fe II AS, Carlo Erba, Milan, Italy) (Na_2_S × 9H_2_O Merck, Milan, Italy), H_2_O_2_ 30% and 2-mercaptoethanol (Merck, Milan, Italy); chloroform, methanol, diethyl ether and *n*-hexane (HPLC grade) were used as received from J. T. Baker, Phillipsburg, NJ, USA. MeOH-d_4_ and DMSO-d_6_ were purchased from Merck (Milan, Italy). NMR spectra were recorded at ambient temperature on a Varian 500 MHz spectrometer (Agilent, Cernusco sul Naviglio, Milan, Italy). Solvents are detailed for each spectrum. UV spectra were recorded at room temperature with a Cary300Bio spectrophotometer (Agilent, Cernusco sul Naviglio, Milan, Italy). Hydrodynamic diameters of the obtained liposomes were measured using the dynamic light scattering (DLS) technique (Malvern Instruments Series NanoZS with a detection angle of 173°, Malvern Instruments, Malvern, UK). All measurements were recorded at 25 °C.

### 2.1. Transesterification Procedure and Gas Chromatography (GC) Analysis

The transesterification protocol optimized for bovine hearth cardiolipin and applied to the phospholipids extracted from liposome suspensions was performed taking care of dry conditions, as follows: 0.5 mg of CL were dissolved in 0.5 mL of a 22 mM NaOMe solution in MeOH corresponding to 32 equiv. in respect of CL, i.e., 8 equiv. in respect of each FA chain (see in Appendix A the details of this preparation) and the resulting mixture was stirred for 2 h at room temperature in a close vessel, under nitrogen atmosphere. Thin-layer chromatography was used to monitor the progress of CL transformation. Eluent: CHCl_3_/MeOH/H_2_O 5:2:0.2, and R_f_ (lyso-CL) = 0.32, R_f_ (CL) = 0.65, R_f_ (18:2 acid) = 0.88, R_f_ (18:2 methyl ester) = 1, to also evidence the hydrolysis of the starting material. After the elapsed time, the reaction mixture was extracted with *n*-hexane (3 × 2 mL) containing 0.1 mL of a standard solution of 17:0 methyl ester; *n*-hexane was then evaporated, the residue dissolved in again in 1 mL of *n*-hexane and 1 µL was injected in GC.

Fatty acid methyl esters (FAME) were analyzed by GC (Agilent 6850, Agilent, Cernusco sul Naviglio, Milan, Italy), using the split mode (50: 1), equipped with a 60 m × 0.25 mm × 0.25 µm Zebron column (Phenomenex, Torrence CA, USA) and a flame ionization detector with the following oven program: temperature started from 165 °C, held for 3 min, followed by an increase of 1 °C /min up to 195 °C, held for 40 min, followed by a second increase of 10 °C /min up to 240 °C, and held for 10 min. A constant pressure mode (29 psi) was chosen with hydrogen as the carrier gas. FAMEs were identified by comparison with authentic samples and chromatograms were examined as described previously. Quantitative studies of the performed reactions were done by multiple-points calibration curves of the standard references (5 points). The GC peak areas were adjusted using a correction factor obtained by the recovery of the 17:0 methyl ester used as standard.

### 2.2. Photolytical Preparation of Mono-Trans CL (MT-CL)

UV irradiations were performed in a micro photochemical reaction assembly with quartz well (Ace Glass Inc., Vineland, NJ, USA) using a 5.5 W cold cathode, low pressure, mercury arc, gaseous discharge lamp (corresponds to λ = 250–260 nm) made of double-bore quartz (Ace Glass Inc., Vineland, NJ, USA). For preparative scope, a higher concentration (3.5 mM) of natural CL is completely dissolved in MeOH. The starting material dissolved in the desired alcohol was added to the photochemical reactor and degassed with N_2_. While degassing, a solution of 2-mercaptoethanol (0.5 equiv.) was added, calculating the volume of the stock thiol solution in order to reach the desired final CL concentration. The reaction mixture was then degassed for 20 min. After the elapsed time the solution was irradiated with at 250–260 nm for the indicated times (4 min for the mono-trans CL, see text). Subsequently, the solvent was evaporated under *vacuum* and the residue was placed under high *vacuum* for 2 h to completely remove 2-mercaptoethanol. Appendix A shows the GC analysis showing the formation of mono-trans isomers. The quantity of MT-CL can be calculated from the ^1^H and ^13^C NMR spectra as explained in the main text. The assignment of NMR signals to MT-CL and other trans-CL isomers (T-CL) are reported in Appendix A and discussed with Figure 3 in Section 3.2. Appendix A shows the four possible structures of MT-CL formed as first products in the isomerization reaction.

### 2.3. Preparative Silver-Thin Layer Chromatography (Ag-TLC)

Preparative TLC glass plates were pre-treated with 5% AgNO_3_ solution in acetonitrile for 15 min and then dried at 120 °C for 1 h [44]. Eluent for separation: CHCl_3_/MeOH/H_2_O 5: 2: 0.2, R_f_ (CL) = 0.52, R_f_ (mix MT-CL) = 0.6, R_f_ (T-CL) = 0.66. The desired spots were scratched off, the silica suspended in absolute EtOH and filtered. Afterwards, the solvent was evaporated under reduced pressure. The Ag-complexes maintained the typical NMR signals with a de-shielding effect due to the presence of metal (see Appendix A).

### 2.4. Preparation of Liposome

Liposomes were obtained by methanolic injection method [45]. Briefly, 1 µmol of phospholipids (POPC or mixture POPC:CL) dissolved in 40 µL of methanol were added in one shot to 960 µL of MQ water (conc. 1 mM) under vigorous stirring with vortex. Stirring was continued at 2200 rpm for 10 min. Then, the suspension was diluted with 1 mL of water to reach a final lipid concentration of 0.5 mM. The characterization of lipid vesicles was performed via DLS.

### 2.5. γ-Radiolysis Experiments

Irradiations were performed at room temperature (22 ± 2 °C) using a ^60^Co-Gammacell at different dose rates. The exact absorbed radiation dose was determined with the Fricke chemical dosimeter, by taking *G*(Fe^3+^) = 1.61 µmol J^−1^ [46]. In the radiation experiments, 4 mL vials equipped with a rubber septum were used. For kinetic experiments, different reactions were prepared from a single stock solution and then each one was stopped at the specific irradiation dose.

A total of 2 mL of freshly prepared stock solutions (0.35 mM) of CL in *i*-PrOH with 18:1 methyl ester (0.33 equiv.) was added together with 2-mercaptoethanol (0.5 equiv.). Afterwards, the solution was degassed with N_2_O for 10 min via a cannula, sealed with parafilm, and irradiated using Gammacell. Transesterification and GC analysis followed the procedure reported above.

A total of 2 mL of freshly prepared stock solutions (0.5 mM) of POPC/CL liposomes in phosphate-buffered saline MQ water (pH = 7.4, POPC/CL ratio 3:1) was added together with 2-mercaptoethanol (1 equiv.). Afterwards, the solution was degassed with N_2_O for 10 min via a cannula, sealed with parafilm, and irradiated. After the irradiation time, the reaction mixtures were worked-up following the procedure described below.

### 2.6. Lipid Extraction from Liposome Suspensions

To the liposome suspensions was added 2: 1 chloroform/methanol solution (5 × 4 mL), according to the Folch method [47]. The organic layer was collected and dried over anhydrous Na_2_SO_4_, then evaporated under vacuum to dryness. The conversion of the extracted phospholipids to the corresponding FAMEs was performed following the transesterification protocol described above. Incubations were carried out in an incubating orbital shaker (ARGOLAB, SKI 4, Carpi, Italy), keeping the temperature at 37 °C.

Large Unilamellar Vesicles (LUV) were prepared using a mixture of 1-palmitoyl-2-oleoyl-*sn*-glycero-phosphatidylcholine (POPC) with bovine hearth CL (stock solution 100 mM, ratio 3: 1). LUV stock suspensions were stored at 4 °C before use. Stock solutions of aqueous Fe^2+^ (1 mM), H_2_O_2_ (1 mM), 2-mercaptoethanol (1 mM), and buffered pH 7.4 Na_2_S (1 mM, pH = 7.4) were freshly prepared before the liposome experiments. To a 2 mL vial were added in sequence: (a) Fe(NH_4_)_2_(SO_4_)_2_ × 6H_2_O (10 µM), H_2_O_2_ (100 µM) liposome (1mM); (b) Fe(NH_4_)_2_(SO_4_)_2_ × 6H_2_O (10 µM), H_2_O_2_ (100 µM), thiol (10 or 100µM), liposome (1 mM). A final reaction volume of 1.6 mL was reached. The reactions were performed by incubation at 37 °C for 15 h followed by work-up as described above (extraction, transesterification and GC analysis).

### 2.7. Incubation Experiments under Oxidative Conditions

Incubations were carried out in an incubating orbital shaker (ARGOLAB, SKI 4, Carpi, Italy), keeping the temperature at 37 °C. Large Unilamellar Vesicles (LUV) were prepared using a mixture of POPC with bovine hearth CL (stock solution 100 mM, ratio 3: 1). LUV stock suspensions were stored at 4 °C before the use. Stock solutions of aqueous Fe^2+^ (1 mM), H_2_O_2_ (1 mM), 2-mercaptoethanol (1 mM), and buffered pH 7.4 Na_2_S (1 mM, pH = 7.4) were freshly prepared before the liposome experiments. To a 2mL vial, the following reagents were added in sequence: Fe(NH_4_)_2_(SO_4_)_2_ × 6H_2_O (10 µM), H_2_O_2_ (100 µM) liposome (1mM); in the presence of thiol the reagents added in sequqnce were: Fe(NH_4_)_2_(SO_4_)_2_ × 6H_2_O (10 µM), H_2_O_2_ (100 µM), thiol (10 or 100µM), liposome (1 mM). In both cases, a final reaction volume of 1.6 mL was reached. The reactions were performed by incubation at 37 °C for 15 h followed by work-up as described above (Section 2.1 and Section 2.6).

## 3. Results and Discussion

### 3.1. Analytical Protocol

Before carrying out the CL isomerization, we examined the transesterification protocol to analyze the FA residues of CL. Either methanolic HCl or methanolic KOH conditions are reported for the lipid transesterification into the corresponding FAME, followed by GC analysis [21,22,23,24]. However, in our hands, the methanolic HCl procedure [21] gave non-reproducible and inconsistent results for CL, due to the decomposition of the starting material. The methanolic KOH procedure, tested at various KOH concentrations, also gave poor results related to the yields, as demonstrated by quantitative evaluation using GC calibration and FA standard references (see Appendix A). In this case, the yield was affected by the hydrolysis of CL to the corresponding free fatty acids, which was never reported with other lipids, such as phosphatidylcholines (PC), under the same conditions [18,48]. As matter of facts, mono-lyso cardiolipins and lyso cardiolipins having 3 and 2 FA chains, respectively, were detected by TLC analysis (see TLC conditions in Appendix A). The yield of the transesterification step became quantitative when strict anhydrous conditions were realized by methanolic NaOMe solution as reagent (prepared as detailed in Appendix A), followed by extraction with *n*-hexane and the removal of the solvent under reduced pressure. The optimization of the reaction conditions resulted in the protocol of 22 mM NaOMe in MeOH and 2 h of magnetic stirring, in a close vessel or under a nitrogen atmosphere, at room temperature (entry 8, Appendix A). Moreover, work-up requires accuracy. In fact, under the NaOMe reaction conditions, inconsistent results were obtained if work-up is performed by simple evaporation of the methanol under vacuum and dissolving the residue in hexane for GC analysis. We also highlighted the use of standard references to check the yields of this analytical protocol at two levels: (*i*) before the extraction step, by the addition of a stock solution of 17:0 methyl ester to the reaction mixture, used as internal standard for the extraction recovery yield and the FAME yields via multiple-points GC calibration curves; (*ii*) by the addition of a stock solution of 19:0 methyl ester as internal standard before performing the transesterification reaction, in order to check if the CL hydrolysis occurs directly on the phospholipid substrate or on the forming methyl esters. The complete recovery of 19:0 methyl ester confirmed that hydrolysis does not occur during our transesterification conditions.

Since cardiolipins are normally present in 20–25% of the total IMM lipids together with other phospholipids, we also tested the optimized method for CL to obtain the simultaneous transformation of a representative phospholipid, POPC, whose transesterification is reported by methanolic KOH [18,48]. Indeed, both lipid classes are effectively transformed into the corresponding FAMEs; therefore, the NaOMe method can be suggested when analysis of biological lipid mixtures is required (see Appendix A).

### 3.2. MT-CL and T-CL Preparation by UV Photolysis

Based on our previous extensive work in thiyl radical-catalyzed cis-trans-isomerization of lipid double bonds [2,16,19,48], we proceeded with the isomerization of natural CL by direct photolysis using 2-mercaptoethanol in MeOH, a solvent where CL is perfectly soluble at high concentrations (3.5 mM). The UV-*vis* absorption spectrum of CL is comparable with methyl linoleate, i.e., both spectra do not show significant absorbance at the wavelength used in the photolysis experiments (λ = 250–260 nm). N_2_-saturated MeOH solution of bovine heart CL (3.5 mM) was irradiated in a micro-photochemical reactor equipped with a high-pressure Hg lamp (*λ* = 250–260 nm) and thiyl radicals are formed according to reactions 1–3 thus starting the CL double bond transformations as shown in Figure 2b.
RSH + *hν* → RS^•^ + H^•^(1)
H^•^ + CH_3_OH → HOCH_2_^•^ + H_2_(2)
HOCH_2_^•^ + RSH → RS^•^ + CH_3_OH (3)

In the first minutes, (4 min) the formation of mono-trans isomers was exclusively obtained and monitored by Ag-TLC, NMR and GC analyses (see Appendix A). The presence of 9trans,12cis-18:2 (**L_EZ_**) and 9cis,12trans-18:2 (**L_ZE_**) moieties in the CL structure is confirmed by GC analysis of the corresponding FAME, using the above described transesterification conditions and known analytical method for isomer separation (Appendix A) [17,18]. The isomerization reaction occurs step-by-step, as shown in Figure 2, i.e., randomly on one of the four fatty acid chains, and the first formed products are likely those containing one trans double bond in one of the linoleoyl chains (Appendix A) shows the four MT-CL structures that can be formed. Another feature of the cis-trans isomerization is that a catalytic amount of thiol can do it, due to the radical chain process that has been fully characterized on linoleoyl residues in previous work [18,19]. In Section 3.4, more details of kinetics will be reported. A non-destructive estimation of the mono-trans CL yield can be obtained using NMR (nuclear magnetic resonance). Figure 3 shows the relevant regions of ^1^H (Figure 3a) and ^13^C (Figure 3b) NMR spectra (full spectra in Appendix A for CL and Appendix A for the CL isomer mix in Appendix A), registered directly on the crude reaction mixture, before transesterification, obtained by evaporating the solvent under reduced pressure and keeping the residue at high vacuum for 2 h to remove residual HO(CH_2_)_2_SH. The crude was dissolved in the deuterated solvent for the analysis. In Figure 3a, the comparison between the ^1^H NMR spectra of natural CL (containing only **L_ZZ_** residues) and the corresponding reaction mix evidences the triplet of the bis-allylic protons at 2.72 ppm and the multiplet of the allylic protons at 1.99 ppm corresponding to the mono-trans structures [19,49]. From the integration of the bisallylic and allylic hydrogen signals, a cis/trans isomer ratio of ca. 6: 1 is envisaged (Appendix A). Taking into account that in the CL isomer structures the trans double-bond-containing chain is present together with the other three cis double-bond-containing chains, under our experimental conditions, the mono-trans CL isomers are formed in ca. 46% yield. An interesting observation came from the ^13^C NMR spectra shown in Figure 3b, focusing on the double bond region. The alkenyl carbon atoms of the natural CL, taking into account previously reported data despite being reported in CDCl_3_ [19,49], are assigned to the C9–C13 carbon atom resonances at a lower field than the C10–C12 carbon atom resonances. For 9cis,12trans-18: 2, the literature reported a 0.7 ppm more de-shielded resonance than in the cis isomer [49]; therefore, we can attribute the peak at 131.7 ppm to the C-13 of the 9cis,12trans-18: 2 -containing CL (see Appendix A for details). By analogy with the CL structure, the signal at 131.6 ppm can be attributed to C-9 in the same molecule. The alkenyl resonances of the other three fatty acid chains of this mono-trans CL likely remain in the same positions than in the natural cis CL. Examining the resonances at 131.3 and 131.2 ppm again, it is possible to attribute to C-9 and C-13 of a second mono-trans CL isomer, by analogy with the 9trans,12cis-18:2 literature data [49]. For the signals at 129.7 and 129.6, 128.8 and 128.7 ppm, the less de-shielded peak at 128.7 ppm is attributed to C-10 of the 9trans,12cis-18:2 followed by the C-12 at 128.8 ppm in the same chain, whereas the resonances at 129.7 and 129.6 ppm individuate the 9cis,12trans-18:2 CL isomer. It is worth noting that another interesting signature of mono-trans CL isomers is the 5 ppm shift of the C-11 (bisallylic) carbon atom, going from 26.04 ppm to 33.61 ppm, as it is known for linoleic acid isomers (Appendix A) [49]. The CL molecule being with a symmetry plane, it is likely that the four mono-trans CL isomers, presenting a 9-trans or a 12-trans double bond in one chain, cannot be distinguished based on whether the trans double-bond-containing chain residue is in an external or internal position with respect to the plane (cf. Appendix A). Overall, the new clearly distinguishable signatures of the ^13^C NMR spectrum represent a milestone, in view of the use of NMR in the “omics” platforms proposed for biological samples.

The GC analysis of the FA residues after 4 min photolysis evidenced the **L_ZZ_**:**L_EZ_**:**L_ZE_**:**L_EE_** ratio to be 84:7.6:7.6:0.8 (see Appendix A). We continued the CL isomerization under photolytic conditions up to 30 min, performing the NMR of the crude (see Appendix A) and the transesterification procedures as described above. We determined the T-CL yield and the isomers’ ratio of **L_ZZ_**:**L_EZ_**:**L_ZE_**:**L_EE_** = 5:13:13:69, which did not change, representing the thermodynamic equilibrium, i.e., the level of isomerization that remains constant over time. It is interesting to observe that, for the analogous isomerization of methyl linoleate, the ratio **L_ZZ_**:**L_EZ_**:**L_ZE_**:**L_EE_** = 4:13:13:70 was determined at thermodynamic equilibrium [19].

### 3.3. Isolation of MT-CL and T-CL as Ag-complexes

Afterwards, we focused on the separation of the trans CL isomers from the reaction crude. The crude mixtures at 4 and 20 min of photolysis were purified employing preparative TLC plates pre-treated with 5% AgNO_3_ solution in acetonitrile, with a previously reported protocol (Ag-TLC) [44]. The desired spots were scratched off and the silica was suspended in absolute EtOH and filtered, thus affording the ethanolic extract. After evaporation under reduced pressure, white solids containing the Ag-complexes of **L_EZ_** + **L_ZE_** CL and **L_EE_** CL were obtained. We also prepared the Ag-complex of the natural CL as reference. In Appendix A, the NMR spectra of the Ag-complexes of CL and T-CL are shown, where it is possible to appreciate the de-shielding of allylic, bis-allylic and alkenyl protons as an effect of the metal complexation of the phospholipid structures. We proceeded with the decomplexation step using saturated NaCl aqueous solution and the forming of an AgCl precipitate. After the lyophilization of this residue treatment with absolute ethanol and filtration, the natural CL and MT-CL isomers were obtained as Na^+^ salt. We observed that the stability of the salt was greatly reduced and the isolated materials were labile. Work is ongoing in our laboratory to have a satisfactory isolation procedure for CL isomers. For the first time, a mix of MT-CL/CL and T-CL/CL can be obtained, knowing the quantity of isomers in the mix, and these modified biomolecules can be used for further studies, as, in our case, we used it to understand the effects on membrane formation, as reported in Section 3.5.

### 3.4. Kinetic and Product Studies by γ-Radiolysis of CL

Thiyl radicals were produced by γ-radiolysis of N_2_O-saturated *i*-PrOH solutions containing CL (0.35 mM) and 2-mercaptoethanol (0.5 equiv.) in the presence of 0.33 equiv. of methyl oleate acting as an internal reference for the isomerization yield. Radiolysis of *i*-PrOH led mainly to solvated electrons e_sol_^−^ and alkyl radicals, as shown in Reaction 4. In the N_2_O-saturated solution, e_sol_^−^ were efficiently quenched to form N_2_ and O^•−^ radical anion (Reaction 5), which could react directly with the solvent to generate (CH_3_)_2_C(^•^)OH (Reaction 6, *k* = 1.9 × 10^9^ M^−1^ s^−1^) [50,51]. Then, the alkyl radicals (R = CH_3_^•^ and (CH_3_)_2_C(^•^)OH) reacted with 2-mercaptoethanol to produce thiyl radicals (Reaction 7).
*i*-PrOH + γ-irr → (CH_3_)_2_C(^•^)OH + e_sol_^−^ + CH_3_^•^(4)
e_sol_^−^ + N_2_O → N_2_ + O^•−^(5)
O^•−^ + *i*-PrOH → (CH_3_)_2_C(^•^)OH + HO^−^(6)
R^•^ + HO(CH_2_)_2_SH → RH + HO(CH_2_)_2_S^•^(7)

We investigated the reaction by detailed products study and dose-dependent experiments, irradiating five samples (0.35 mM CL each) for 100 Gy dose intervals, until reaching the total dose of 400 Gy in the last sample. The reaction mixtures were worked-up and analyzed as described in Appendix A. The results shown in Table 1 depict the extent of the isomerization of the two major unsaturated fatty acid residues from CL (linoleic and vaccenic acids), and the addition of methyl oleate (0.33 mM) was used as an internal reference for the isomerization course. Appendix A shows the corresponding graphs of the reaction course. While the isomerization of the vaccenic residue and methyl oleate proceeded proportionally to the dose course, the extent of the isomerization of linoleic acid residues showed a more complex pattern. Initially, the step-by-step isomerization process generated the two mono-trans isomers with the same percentages (100 Gy, 13% each in Table 1). Proceeding with the dose, the levels of the mono-trans remained constant at values of ca. 22%, while the formation of **L_EE_** increased gradually, achieving 32.8% at 400 Gy dose.

Figure 4 displays the reaction course for the isomer concentrations as a function of irradiation dose for 0.35 mM of CL, which corresponds to 1.40 mM of **L_ZZ_** residues, and 0.5 equiv. HO(CH_2_)_2_SH. The diminution of **L_ZZ_** matched with the formation of mono-trans and di-trans isomers (**L_ZE_**, **L_EZ_**, and **L_EE_**). The two mono-trans isomers were found in equal amounts and are reported together in Figure 4. However, the sum of all geometrical isomers (total **L**) proportionally decreased by increasing the dose (ca. 20%), indicating secondary products formation to account for the full mass balance. It is gratifying to see the similarity of this figure with the analogous radiolysis experiment of methyl linoleate [19]. The diminution of total **L** can be probably attributed to hydrogen abstraction from the bis-allylic position by thiyl radicals and adduct formation with thiol, as previously established for methyl linoleate [19].

The disappearance of the starting material (mol kg^−1^) divided by the absorbed dose (1 Gy = 1 J kg^−1^) gives the radiation chemical yield (*G*) or *G*(–**L_ZZ_**). The extrapolation to zero dose gives G = 5.06 μmol J^−1^ (see Appendix A). Assuming that the *G*(RS^•^) is 0.65 μmol J^−1^, we calculated the catalytic cycle to be 8 at the initial phase. For the analogous isomerization of 4.7 mM methyl linoleate and 20 mM HO(CH_2_)_2_SH, a catalytic cycle of 13 was calculated [19].

### 3.5. Large Unilamellar Vesicles (LUV) Containing CL, MT-CL and T-CL

Next, we examined the behavior of cardiolipins in the supramolecular organization of LUV [48]. Biomimetic models of LUV containing 3: 1 POPC:CL (the cardiolipin percentage typically present in IMM) were prepared using POPC in a 3: 1 mixture with MT-CL or T-CL (obtained, as above described, after 4 min and 20 min photolysis of CL, respectively). LUV were obtained by the methanolic injection method [45] (as described in Appendix A). The characterization of the resulting LUVs was performed via dynamic light scattering (DLS), and the average diameters are summarized in Table 2, in comparison with liposomes prepared only with POPC. The presence of CL resulted in an increase in the hydrodynamic diameter from 93.2 to 198 nm (entry 2, Table 2); it is interesting to see that, using lipid mixtures containing MT-CL/CL and T-CL/CL, a contraction of the vesicle diameter of ca 16% and 26%, respectively, was observed (cf. entries 3 and 4, Table 2). This result evidences that the geometrical trans isomer CL structure determines a tighter packing of the lipid layer than the cis one, affording a decrease in the vesicle size. This effect of reduced diameter, due to the different arrangement of trans fatty acid residues, was previously observed in phospholipid vesicles containing the 9trans-18:1 moiety [33].

With the cardiolipin-containing liposomes in our hands, we progressed our investigation on their reactivity with thiyl radicals generated by either by γ-radiolysis or Fenton-type reaction in aqueous solution.

### 3.6. γ-Radiolysis of CL-Containing Liposomes

The radiolysis of neutral water leads mainly to the reactive species e_aq_^−^, HO^•^, and H^•^, together with H^+^ and H_2_O_2_, as shown in Reaction 8. The values in brackets represent the radiation chemical yield (*G*) in µmol J^−1^ [51]. In N_2_O-saturated solution (≈ 0.02 M of N_2_O), e_aq_^−^ are efficiently transformed into HO^•^ radicals via Reaction 9 (*k* = 9.1 × 10^9^ M^−1^s^−1^), affording G (HO^•^) = 0.55 µmol J^−1^; that is, HO^•^ and H^•^ account for 90% and 10%, respectively, of the reactive species. Subsequently, HO^•^ and H^•^ atoms react with RSH, affording the desired thiyl radicals (Reactions 10 and 11, *k* = 6.8 × 10^9^ M^−1^s^−1^ and *k* = 1.7 × 10^9^ M^−1^s^−1^, respectively) [50,51].
H_2_O + γ-irr → e_aq_^−^ (0.27), HO^•^ (0.28), H^•^ (0.06) (8)
e_aq_^−^ + N_2_O + H_2_O → HO^•^ + N_2_ + HO^−^(9)
HO^•^ + RSH → H_2_O + RS^•^(10)
H^•^ + RSH → H_2_ + RS^•^(11)

Five independent liposome samples, containing 3: 1 POPC/CL, in water (0.5 mM), in the presence of 0.5 mM HO(CH_2_)_2_SH, were used in a dose-dependent experiment. The amount of thiol was 3.3-fold higher than in solution in order to properly evaluate the reactivity of linoleic acid residues. Each sample was stopped after 100 Gy dose intervals until reaching a total dose of 400 Gy (Table 3, see also Appendix A). The isomerization of vaccenic and oleic acid moieties proceeded proportionally to the dose course, similar to the above-described isomerization in *i*-PrOH solutions. Looking at the 100 Gy dose experiment, the outcome clearly highlighted a different reactivity of **L_ZZ_** residues in liposomal organization. Interestingly, while in solution, the two mono-trans isomers were generated at the same extent (cf., Table 1); in this case, we observed that the preferential formation of **L_EZ_**, ca. was 6% higher than **L_ZE_**. The formation of di-trans linoleate residues (**L_EE_**) reached 11.3%, 5 times higher than the corresponding time point in the previous experiment in solution. Subsequently, at 200 Gy, the levels of **L_EZ_** and **L_ZE_** reached a plateau at ca. 18 and 11%, respectively, and remained constant, while the amount of **L_EE_** increases progressively. Figure 4 shows the concentration of each isomer (mM) as a function of the irradiation dose, including the total **L** (the sum of all geometrical isomers). The different amounts of mono-trans isomers and di-trans isomers in the liposome experiment are explained by the diffusion of the amphiphilic HOCH_2_CH_2_S^•^ radical into the membrane assembly, and the supramolecular organization of the lipid bilayer. In fact, this radical freely diffuses between aqueous and lipid compartments and, when it goes through the hydrophobic region of the membrane bilayer, the double bonds of the CL structures closest to the glycerol moiety are the first to be reached and react. Therefore, the reactivity outcome is driven by both the supramolecular arrangement of the hydrocarbon tails and the highly defined lateral diffusion. Due to the packing of fatty acid residues and, consequently, to a highly defined lateral diffusion, thiyl radicals, after the isomerization of the 9,10-double bond, have a chance to isomerize the second double bond in the same chain before migrating laterally. The present results are in accordance with the previously reported PUFA reactivity of, e.g., linoleic and arachidonic acids [2,48]. This “positional effect” can be investigated in further work on the endogenous formation of trans lipids in mitochondrial membranes.

As depicted in Figure 5, the initial amount of methyl linoleate (purple line) proportionally decreased by increasing dose (ca. 50%), compared to the experiment in solution, which is larger than the experiment in solution (ca. 20%) due to the higher amount of thiol used. Although we did not identify products other than geometrical isomers in this experiment, based on our previous publications, the reactivity and products derived from hydrogen abstraction from the bis-allylic position by thiyl radicals and to adducts with thiol are likely to occur [19].

Regarding the radiation chemical yield (*G*), the plots vs. dose is reported in Appendix A. The extrapolation of G at zero dose gives: *G* (–**L_ZZ_**) = 2.4 µmol J^−1^, *G* (**L_EZ_**) = 0.8 µmol J^−1^, *G* (**L_ZE_**) = 0.5 µmol J^−1^, and *G* (**L_EE_**) = 0.6 µmol J^−1^ (Appendix A). Assuming that the *G*(RS^•^) is 0.65 μmol J^−1^, we calculated the catalytic cycle to be 4 at the initial phase. Moreover, the *G* for the disappearance of the sum of geometrical isomers is 0.6 µmol J^−1^ (Appendix A), suggesting that each thiyl radical consumes one molecule of **L_ZZ_** or its geometrical isomer.

Additionally, each irradiated solution was analyzed via dynamic light scattering (DLS) prior to the work-up. The measured vesicle diameters are summarized in Table 4. It was gratifying to observe that the size of the lipid vesicles decreased by ca. 10–15% as long as the cis-trans isomerization proceeds (Table 4). The detected diameters decreased with the progressive double bond isomerization and the amounts of trans-containing fatty acid chains (cf., Table 2). Although this model does not represent the lipid diversity of natural mitochondria, it effectively shows the consequences of the trans geometry produced in the structural organization of membranes.

### 3.7. CL-Containing Liposomes Under Oxidative Conditions.

In order to assay free radical reactivity leading to cardiolipin isomerization under conditions of oxidative stress [15], experiments were carried out under biologically related conditions, i.e., in the presence of oxygen and under the free radical conditions created by Fenton reagents. A total of 1 mM of aqueous liposome suspension composed of 3: 1 POPC/CL was incubated at 37 °C, without degassing and leaving the vial under air for 15 h (“open air”), in the absence or presence of thiols, i.e., choosing two different amphiphilic thiols, such as HO(CH_2_)_2_SH and H_2_S. The latter was generated from its sodium salt Na_2_S, used at two different concentrations (10 and 100 µM), and at the same time in the presence of Fe^2+^ salt and 100 µM H_2_O_2_ (see Experimental details as reported in Appendix A). The reaction of the reduced-state transition metal ions, such as Fe^2+^, with H_2_O_2_ is known to give HO^•^ radicals according to the reaction 12 [52,53].
H_2_O_2_ + Fe^2+^ → HO^•^ + HO^−^ + Fe^3+^(12)

This reaction, referred to as the Fenton reaction, has been reported to be responsible for some of the toxicity associated with H_2_O_2_ in vivo, and is certainly characteristic of the mitochondrial chemistry. The combined presence of thiols maintains iron in the reduced state (Fe^2+^) with the formation of thiyl radical (reaction 13):RSH + Fe^3+^ → RS^•^ + Fe^2+^(13)

Both HO^•^ and RS^•^ are able to abstract the bisallylic hydrogen from LH and generate the bisallylic radical in the linoleic residue (reaction 14) that starts the radical chain lipid peroxidation (reactions 15 and 16), whereas the cis-trans isomerization of double bonds of oleic and linoleic acids occurs by RS^•^ radicals [20,54].
HO^•^/ RS^•^ + LH → H_2_O/RSH + L^•^(14)
L^•^ + O_2_ → LOO^•^(15)
LOO^•^ + LH → LOOH + L^•^(16)

The results are shown in Table 5 reporting the transformations of the two main unsaturated fatty acid moieties composing POPC and CL, i.e., oleic acid (9cis-18:1) and linoleic acid (**L_ZZ_**), respectively, as obtained from triplicates of the reaction under the indicated conditions. The cis-trans isomerization was present with both thiols and at both concentrations, with small variations that are not worth being considered at this point. Without thiol, the isomerization did not occur.

Under incubation without the removal of oxygen and in the presence of Fenton reagent, it can be expected that linoleic acid residue is consumed by oxidative pathways. In our experiments, we did not directly measure the oxidative by-products, but we indirectly evaluated the linoleic acid loss or consumption, by quantifying the remaining linoleic acid residues by calibration and an appropriate internal standard added to the reaction mixture (C17:0). It is interesting to note that, since both oxidation and isomerization reactions occur, this means that, under the conditions used, thiol compounds are not able at the concentrations used to protect the unsaturated lipids from oxidative degradation under free radical conditions.

We must underline the fact that, in our experiments, we kept the molarities of lipids and thiols (0.5–1 mM and 0.35 µM, respectively), as well as the nanomolar generation of radical species, closest to the biological ones. The results obtained with the POPC/CL mix similar to the mitochondrial ratio, with and without oxygen, indicate that both lipids can react under free radical conditions, with variable amounts of trans lipids formed depending on the conditions used. However, our models cannot straightforwardly reproduce the complexity of a real cellular environment.

## 4. Conclusions

The thiyl radical-catalyzed cis-trans isomerization of double bonds is a flexible methodology to achieve non-natural trans fatty acid derivatives of great importance for multidisciplinary aspects going from chemistry to molecular biology and medicine. Our results clarify the chemical reactivity under radiolysis and photolysis conditions of an important biomolecule, such as CL, involved in the function of mitochondria. We provide the first synthetic procedure for trans cardiolipin isomers that can be useful for experiments defining their biological effects. Our careful chemical analytical study provides important insights into protocols to identify trans CL isomers, highlighting tools such as GC and NMR and creating awareness on the mass data information for mitochondria evaluation, which cannot distinguish CL isomers having the same molecular mass [25,26]. Our protocols draw more attention for the work-up of biological samples required in lipidomics and metabolomics research.

Our results in the catalytic cycle of CL isomerization in solution and liposomes induced by thiyl radicals and the results obtained in biomimetic models under oxidative conditions can have a realistic application in biological systems, taking into account the molarity of lipids and different thiols in organisms. Our present data are limited to model experiments that must have confirmation for their application to biological systems. Two scenarios relevant for our data can be depicted: (i) the presence of H_2_S, at remarkable concentrations after its enzymatic production in mitochondria [55,56], and the isomerizing ability of its small diffusible radicals, such as HS^•^ or the deprotonated form S^•−^ [57], suggesting to monitor mitochondria with in vitro and in vivo experiments; (ii) the CL oxidation and the effects of several protective agents, monitored by the increase in CL contents [24], indicating the importance of a follow-up of CL isomerization under various stress conditions, in order to individuate antioxidants able to also preserve the natural cis double bond geometry [28]. The need for a multidisciplinary approach in the study of this interesting biomolecule clearly emerges.

## Figures and Tables

**Figure 1 biomolecules-10-01189-f001:**
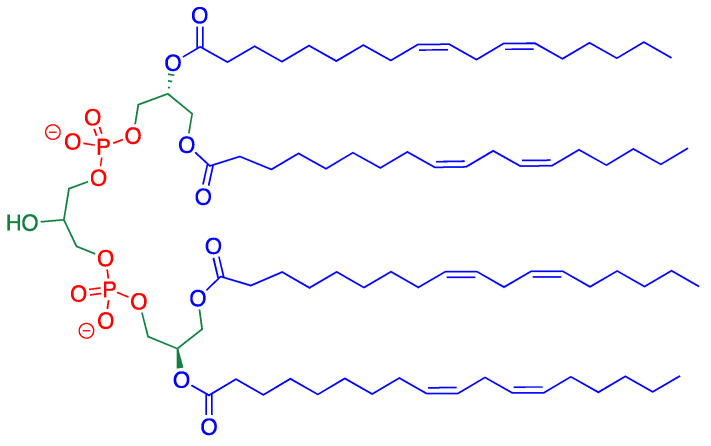
Structure of tetra-linoleoyl cardiolipin.

**Figure 2 biomolecules-10-01189-f002:**
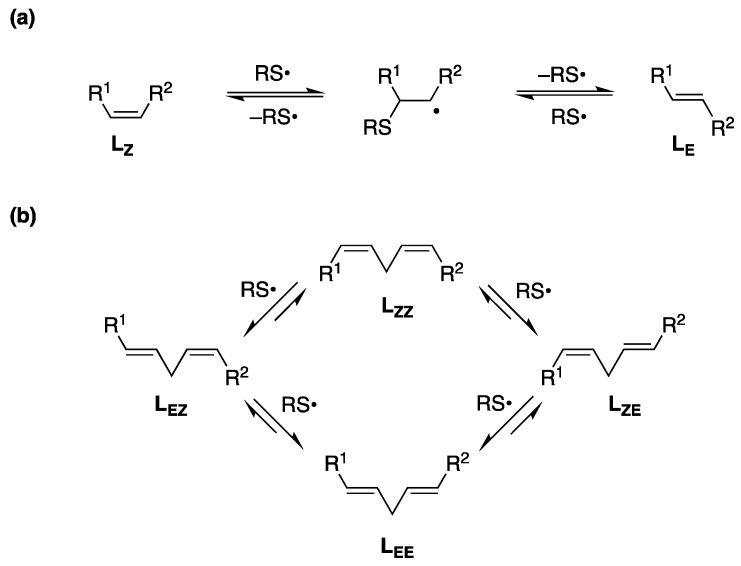
(**a**) Thiyl radical-catalyzed isomerization of monounsaturated fatty acid moiety **L_Z_** to **L_E_**; (**b**) Cis−trans isomerization of linoleic moiety (**L_ZZ_**) catalyzed by thiyl radicals.

**Figure 3 biomolecules-10-01189-f003:**
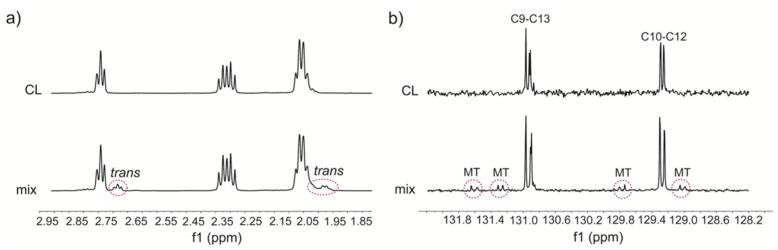
NMR spectral regions of natural cardiolipin (CL) and its reaction crude in the first minutes of UV photolysis containing mono-trans CL (MT-CL, see Appendix A for the structures): (**a**) ^1^H NMR region relative to bis-allylic and allylic proton signals; (**b**) ^13^C NMR region relative to alkene carbon atom signals.

**Figure 4 biomolecules-10-01189-f004:**
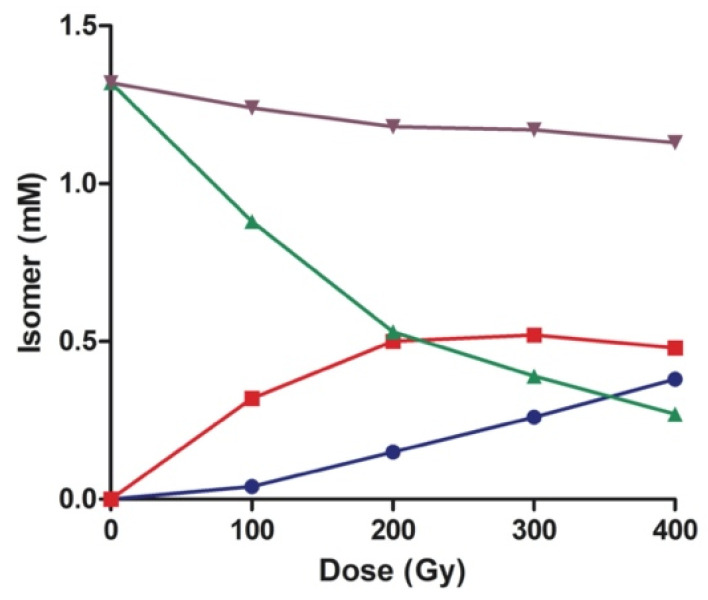
Dose profile of cis−trans isomerization of linoleoyl residue (**L_ZZ_** corresponding to 1.32 mM) of CL (0.35 mM) catalyzed by thiyl radicals generated by γ-radiolysis of HO(CH_2_)_2_SH (0.5 equiv.) in the presence of 0.33 equiv. of methyl oleate as control in N_2_O-saturated *i*-PrOH at 22 °C. Profiles of **L_ZZ_** (green ▲), **L_ZE_** + **L_EZ_** (red ■) and **L_EE_** (blue ●), and the sum of all geometrical isomers (purple ▼).

**Figure 5 biomolecules-10-01189-f005:**
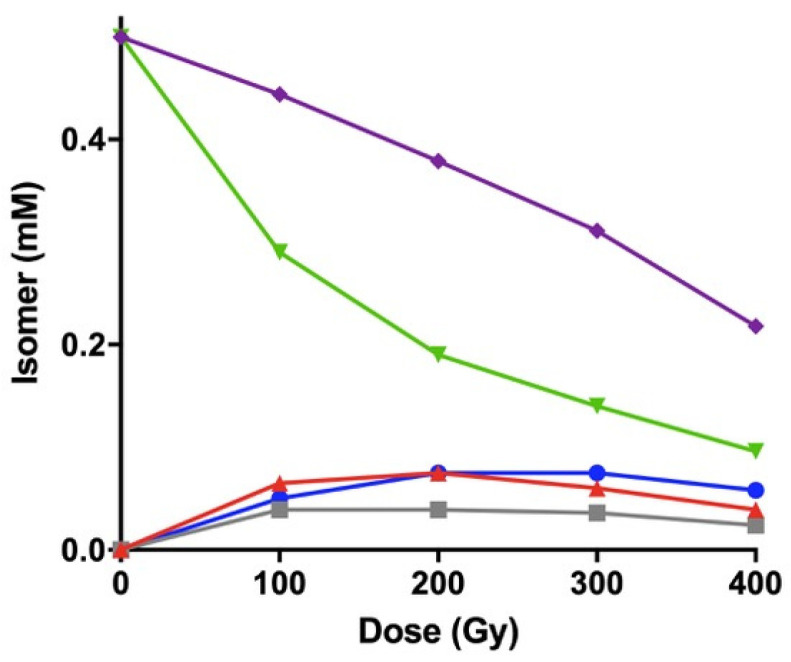
γ-Radiolysis of 0.5 mM POPC/CL liposomes (3:1) in the presence of 0.5 mM HO(CH_2_)_2_SH as a function of the irradiation dose. The concentration (mM) of **L_ZZ_** (green ▼), **L_ZE_** (grey ■), **L_EZ_** (red ▲) and **L_EE_** (blue ●), and the sum of all geometrical isomers (purple ◆).

**Table 1 biomolecules-10-01189-t001:** γ-Radiolysis of CL (0.35 mM) and HO(CH_2_)_2_SH (0.5 equiv.) in N_2_O-saturated *i*-PrOH at 22 °C.

Dose, Gy	Internal Control (% rel) ^1,2^		CL Fatty Acid Residues (% rel) ^2^					
	9cis-18:1	9trans-18:1	11cis-18:1	11trans-18:1	L_ZZ_	L_EZ_	L_ZE_	L_EE_
0	100	0	100	0	100	0	0	0
100	84.4	15.6	83.7	12.3	71.3	13.0	13.0	2.7
200	67.2	32.8	64.9	35.1	44.9	21.2	21.2	12.7
300	53.4	46.6	55.6	44.4	33.8	22.1	22.2	21.9
400	40.5	59.5	44.9	55.1	24.6	21.4	21.2	32.8

^1^ Methyl oleate (9cis-18:1; 0.33 equiv.) as internal standard. ^2^ After the transesterification of reaction crudes (yields > 95%), the ratios of geometrical isomers were determined by GC analysis as relative percentages (% rel) of isomers, calculated separately for the two 18:1 and 18:2 residues of CL.

**Table 2 biomolecules-10-01189-t002:** Dynamic light scattering (DLS) analysis of liposomes (1 mM) in phosphate-buffered water (pH = 7.4) of different 1-palmitoyl-2-oleoyl phosphatidylcholine (POPC) and CL compositions.

Entry	Phospholipid ^1^	Ratio	Diameter, nm	Polydispersity
1	POPC	-	93.2	0.211
2	POPC/CL	3: 1	198	0.278
3	POPC/MT-CL ^2^	3: 1	167	0.151
4	POPC/T-CL ^3^	3: 1	146.1	0.394

^1^ 1mM; ^2^ MT-CL obtained after 4 min photolysis of CL. ^3^ T-CL obtained after 20 min photolysis of CL.

**Table 3 biomolecules-10-01189-t003:** γ-Radiolysis of POPC/CL liposomes (0.5 mM) in N_2_O-saturated phosphate-buffered water (pH = 7.4) and HO(CH_2_)_2_SH (1 equiv.) at 22 °C.

Dose, Gy	18: 1 from POPC (% rel) ^1^		CL fatty acid residues (% rel) ^1^					
	9cis-18:1	9trans-18:1	11cis-18:1	11trans-18:1	L_ZZ_	L_EZ_	L_ZE_	L_EE_
0	100	0	100	0	100	0	0	0
100	65.4	34.6	70.9	29.1	65.3	14.6	8.8	11.3
200	50.7	49.3	56.2	43.8	51.4	17.8	10.7	20.1
300	38.8	61.2	49.2	50.8	44.5	18.5	11.5	25.5
400	34.6	65.4	43.0	57.0	44.2	17.9	11.1	26.8

^1^ After the transesterification of reaction crudes (yields > 95%), the geometrical isomers ratios were determined by GC analysis as relative percentages (% rel) of isomers, calculated separately for 18:1 and 18:2 residues.

**Table 4 biomolecules-10-01189-t004:** DLS analysis of irradiated POPC/CL liposomes (0.5 mM) in phosphate-buffered water (pH = 7.4).

Dose, Gy	Diameter, nm	Polydispersity
0	198	0.278
100	187.4	0.310
200	184.4	0.286
300	184	0.303
400	176.6	0.279

**Table 5 biomolecules-10-01189-t005:** POPC/CL liposome (1 mM) Suspension in Phosphate-Buffered Water (pH = 7.4) in the Presence of 10 µM Fe^2+^, 100 µM H_2_O_2_ and Thiol, Incubated at 37 °C in Open Air for 15 h.

Thiol, µM	18:1 from POPC (% rel) ^1^		18:2 from CL^1^		
	9cis-18:1 ^2,3^	9trans-18:1 ^2,3^	L_ZZ_ ^3^	L_EZ_ + L_ZE_ ^3^	L_ZZ_ Consumption ^3^
-	100	-	14.5 ± 0.9	-	85.0 ± 1.2
HO(CH_2_)_2_SH, 10	99.6	0.4	70.0 ± 1.4	0.6 ± 0.1	29.4 ± 1.4
HO(CH_2_)_2_SH, 100	99.0	1.0	48.2 ± 1.3	3.5 ± 0.4	48.3 ± 1.3
H_2_S, 10	99.7	0.3	63.6 ± 1.2	0.2 ± 0.1	36.2 ± 1.2
H_2_S, 100	99.1	0.9	44.5 ± 3.3	0.6 ± 0.1	54.9 ± 3.3

^1^ The values are means ± standard deviation of *n* = 3 experiments under the same conditions; ^2^ Errors < 0.1; ^3^ Values are expressed as a relative percentage (% rel) of each fatty acid isomer with respect to the starting cis fatty acid residue, estimated using calibration and 17:0 (heptadecanoic acid methyl ester) as an internal standard in the reaction mixture, after the transesterification of the fatty acid moieties composing POPC and CL and their quantification by GC.

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
