# Peer review of "Free-Radical-Mediated Formation of Trans-Cardiolipin Isomers, Analytical Approaches for Lipidomics and Consequences of the Structural Organization of Membranes†"

_biomolecules, 2020, doi:10.3390/biom10081189_

Round 1
Reviewer 1 Report
Dear Editor,
I have read with interest the manuscript presented by Dr. Chatgilialoglu and Dr. Ferreri (and collaborators) on the Free Radical-Mediated Formation of Trans Cardiolipin isomers.
In particular, the Authors investigate the cis-trans isomerization of unsaturated fatty acids – constituting the CL side chains - induced by thiyl radicals (RS.).
The study is quite well done. I appreciated the order and the explanations of the investigation and its accuracy. Surely the study will be useful to other researchers in this field and related field. The SI-file complements the manuscript.
Actually I have only few minor observations that could help the Authors being more specific in their comments and put the study in the context.
Minor
1) Lines 220-234: the Authors report a very relevant observation, often neglected by others while discussing the phospholipid hydrolysis for FA determination. I am very satisfied by reading this critical note, but I would like to ask the authors to briefly comment, on the manuscript, about the motivations behind the observed lack of consistency for acid or basic hydrolysis. Thank you.
2) Line 254 and all other points where this discussion is relevant. It is very important that the Author critically discuss the importance of the thiol:alkene molar ratio on the isomerization process, because this imply how much realistic is the scenario here presented. What are the strict requirements in terms of thiol:alkene molar ratio? Is there any realistic counterpart in cellular conditions?
3) Line 315 and following. Please discuss whether the experimental conditions (continuous irradiation for 4-20 minutes, or the dose in Table 1) is a realistic model for in vivo transformations, and if not, please comment on that.
4) section 3.6: I am not sure whether I have well understood: what can be said about the intrinsic reactivity of unsaturated chains of POPC vs CL? Please state clearly the conclusions from this study.
5) can the Author comment on the possible use of cysteine-containing amino acids or peptides as realistic thiols instead of H2S or beta-mercaptoethanol?
6) Authors should state, even roughly, what is the error bar associated to their measurements, for example by reporting a typical between-experiment variations (as an order of magnitude).
7) it is generally a good suggestion to highlight, in the conclusions, also the limitations of current approach and sketch what could be the next steps for better understanding this very interesting system.
Typos
-----
- 1) Line 116: maybe the capital O of oleate is not needed (?)
- 2) line 118, 213, 215: the O of H2O looks like a zero (0)
- 3) line 198: maybe the capital C of cardiolipin is not needed
- 4) line 197-198: the name of POPC differ from line 116. Please use a uniform naming
- 5) line 116-117: The “sn” should be rendered in italics.
- 6) Line 298: is “metabolomics” here used for “lipidomics”?
Author Response
Answers to Reviewer # 1
Minor
- Lines 220-234: the Authors report a very relevant observation, often neglected by others while discussing the phospholipid hydrolysis for FA determination. I am very satisfied by reading this critical note, but I would like to ask the authors to briefly comment, on the manuscript, about the motivations behind the observed lack of consistency for acid or basic hydrolysis. Thank you.
Thank you for this observation and request: indeed, we found that hydrolysis due to presence of water (also in traces) can affect this transformation step and our conditions using NaOMe were accurately determined. Following the request of this referee, in the main text line 224-239 were modified. Also, in Experimental section (2.1) we added the specific conditions of transesterification and we added in the SM we added the details of the preparation of the NaOMe to ensure dry conditions.
- Line 254 and all other points where this discussion is relevant. It is very important that the Author critically discuss the importance of the thiol:alkene molar ratio on the isomerization process, because this imply how much realistic is the scenario here presented. What are the strict requirements in terms of thiol:alkene molar ratio? Is there any realistic counterpart in cellular conditions?
This observation from the referee prompted us to slightly modify the description of the mechanism (Fig 2) given in lines 270-273 to evidence that thiyl radicals react with lipid double bonds in a catalytic fashion, and the catalytic cycles are of different length according to several factors. Indeed, in lines 180-182 and 464-465 we estimated the chain lengths of 8 and 4 in i-PrOH and in aqueous solutions, respectively. We estimated the conditions used in this work to be “biologically related” but they cannot be straightforwardly applied to a biological scenario which is more complex. We added lines 519-522 to conclude this “realism”.
- Line 315 and following. Please discuss whether the experimental conditions (continuous irradiation for 4-20 minutes, or the dose in Table 1) is a realistic model for in vivo transformations, and if not, please comment on that.
The 4 and 20 minutes-irradiations were made for synthetic purposes, i.e., for the synthesis of the mono-trans and di-trans cardiolipins, also using a concentration of thiol 10 times higher than in the biologically related mechanistic experiments. In lines 334-336 we added that this is a synthetic part that allow for the availability of these modified biomolecules.
- section 3.6: I am not sure whether I have well understood: what can be said about the intrinsic reactivity of unsaturated chains of POPC vs CL? Please state clearly the conclusions from this study.
We used the natural CL (from bovine hearth) that contains mainly linoleic acid and 5% of vaccenic acid. The mix POPC/CL was made by creating a 3/1 mix that realizes the 25% of CL presence in the IMM cristae. In lines 520-522 we cited that the data of POPC/CL mixtures confirms that, using biologically related proportions between the two lipids, the isomerization can occur with different trans isomer formation depending on the conditions used.
5) can the Author comment on the possible use of cysteine-containing amino acids or peptides as realistic thiols instead of H2S or beta-mercaptoethanol?
In the present work we wanted to focus on the diffusible radicals, such as mercaptoethanol and H2S. We mentioned, in the Introduction (lines 99-101) and ref 2, our work with cysteine-containing amino acids and peptides but in the comments reported in the conclusions (Lines 547-549) we believe that the most connected process in mitochondria is the production of radical species from H2S.
6) Authors should state, even roughly, what is the error bar associated to their measurements, for example by reporting a typical between-experiment variations (as an order of magnitude).
We added the errors <5% under Tables 1-3. The errors are combined with the transesterification procedures (such errors are reported in Table S2).
7) it is generally a good suggestion to highlight, in the conclusions, also the limitations of current approach and sketch what could be the next steps for better understanding this very interesting system.
We followed the advice of the referee (line 522-523 and line 545-546)
Typos
-----
- 1) Line 116: maybe the capital O of oleate is not needed (?) corrected
- 2) line 118, 213, 215: the O of H2O looks like a zero (0) corrected
- 3) line 198: maybe the capital C of cardiolipin is not needed corrected
- 4) line 197-198: the name of POPC differ from line 116. Please use a uniform naming corrected
- 5) line 116-117: The “sn” should be rendered in italics. corrected
- 6) Line 298: is “metabolomics” here used for “lipidomics”? line 305 corrected the phrase
Reviewer 2 Report
In the present study, the authors establish a methodological framework for the detection of trans-cardiolipin isomers (T-CLs) and show the formation of T-CLs upon UV-photolysis, gamma-radiolysis or under oxidative conditions. To my knowledge, this is the first study proving the formation of T-CLs experimentally. The procedures described are technically sound and seemingly applicable for the lipid extracts from biological samples. Although the method development may already merit publication in Biomolecules, following points should be considered prior to publication.
Specific points
- The conditions used to generate T-CLs were relatively harsh and might not happen in physiological circumstances. Moreover, the authors used only pure synthetic lipids in their studies. In my view, the present study does not provide any conclusive evidence for the formation of T-CLs in mitochondria in living cells. The statement in abstract “These results highlight a new molecular modification pathway of mitochondrial lipids” should be toned down to avoid misunderstanding that CL isomerization is proven to occur in mitochondria. Along with this line, the drawing of mitochondria at the bottom left corner in Graphical Abstract should also be removed. When the authors aim to point physiological relevance of their study, they must provide an evidence for the formation of T-CLs in a biological sample (e.g. in isolated mitochondria from cells treated with a chemical inhibitor of OXPHOS). As the authors discussed, formation of T-CLs in mitochondria is certainly possible, and this study opens up a possibility to analyze it. I appreciate it very much. However, misleading should be avoided.
- The authors conclude that T-CLs determine tighter packing of lipid layer (L394). However, DLS analysis of liposomes does not measure membrane packing status on them directly. The analysis should be corroborated by another method, for example by monitoring fluorescent change of polarity sensitive probes or by molecular dynamics simulations.
Author Response
Answers to Reviewer # 2
Specific points
- The conditions used to generate T-CLs were relatively harsh and might not happen in physiological circumstances. Moreover, the authors used only pure synthetic lipids in their studies. In my view, the present study does not provide any conclusive evidence for the formation of T-CLs in mitochondria in living cells. The statement in abstract “These results highlight a new molecular modification pathway of mitochondrial lipids” should be toned down to avoid misunderstanding that CL isomerization is proven to occur in mitochondria. Along with this line, the drawing of mitochondria at the bottom left corner in Graphical Abstract should also be removed. When the authors aim to point physiological relevance of their study, they must provide an evidence for the formation of T-CLs in a biological sample (e.g. in isolated mitochondria from cells treated with a chemical inhibitor of OXPHOS). As the authors discussed, formation of T-CLs in mitochondria is certainly possible, and this study opens up a possibility to analyze it. I appreciate it very much. However, misleading should be avoided.
We amended all places where the misleading meaning of our words could occur. We changed the drawing in the Graphical Abstract. In lines 542-552 we clarified the place of our work in view of further biological experiments.
- The authors conclude that T-CLs determine tighter packing of lipid layer (L394). However, DLS analysis of liposomes does not measure membrane packing status on them directly. The analysis should be corroborated by another method, for example by monitoring fluorescent change of polarity sensitive probes or by molecular dynamics simulations.
The DLS experiment was performed to show the different organization of the vesicles in terms of diameter. We clearly mentioned this property and did not extrapolate to other properties that, as the referee correctly points, need specific experiments such as detection of fluorescence changes and corroboration with techniques such as dynamic simulation. This will be matter of further research that we believe will attract specialists in these fields.